Rates and modes of body size evolution in early carnivores and herbivores: a case study from Captorhinidae

Brocklehurst Neil neil.brocklehurst@mfn-berlin.de
Leibniz-Institut für Evolutions- und Biodiversitätsforschung, Museum für Naturkunde , Berlin , Germany
Wilson Laura
Electronic publication date: 2016 Jan 11
Publication date: 2016
Volume: 4
Electronic Location ID: e1555
Received 2015 Sep 17; Accepted 2015 Dec 11
Copyright: © 2016 Brocklehurst
Copyright year: 2016
Copyright holder: Brocklehurst
License: This is an open access article distributed under the terms of the Creative Commons Attribution License, which permits unrestricted use, distribution, reproduction and adaptation in any medium and for any purpose provided that it is properly attributed. For attribution, the original author(s), title, publication source (PeerJ) and either DOI or URL of the article must be cited.
License URL: https://creativecommons.org/licenses/by/4.0/

Keywords: Captorhinidae, Body size, Rate, Evolution, Diet, Herbivore

Funding: This study was funded by a Sofja Kovalevskaja Award to Professor Jörg Fröbisch of the Museum für Naturkunde, which is awarded by the Alexander von Humboldt Foundation and donated by the German Federal Ministry for Education and Research. The funders had no role in study design, data collection and analysis, decision to publish, or preparation of the manuscript.

==============================
Body size is an extremely important characteristic, impacting on a variety of ecological and life-history traits. It is therefore important to understand the factors which may affect its evolution, and diet has attracted much interest in this context. A recent study which examined the evolution of the earliest terrestrial herbivores in the Late Carboniferous and Early Permian concluded that in the four herbivorous clades examined there was a trend towards increased body size, and that this increase was more substantial than that observed in closely related carnivorous clades. However, this hypothesis was not based on quantitative examination, and phylogenetic comparative methods provide a more robust means of testing such hypotheses. Here, the evolution of body size within different dietary regimes is examined in Captorhinidae, the most diverse and longest lived of these earliest high fibre herbivores. Evolutionary models were fit to their phylogeny to test for variation in rate and mode of evolution between the carnivorous and herbivorous members of this clade, and an analysis of rate variation throughout the tree was carried out. Estimates of ancestral body sizes were calculated in order to compare the rates and direction of evolution of lineages with different dietary regimes. Support for the idea that the high fibre herbivores within captorhinids are being drawn to a higher adaptive peak in body size than the carnivorous members of this clade is weak. A shift in rates of body size evolution is identified, but this does not coincide with the evolution of high-fibre herbivory, instead occurring earlier in time and at a more basal node. Herbivorous lineages which show an increase in size are not found to evolve at a faster rate than those which show a decrease; in fact, it is those which experience a size decrease which evolve at higher rates. It is possible the shift in rates of evolution is related to the improved food processing ability of the more derived captorhinids rather than a shift in diet, but the evidence for this is circumstantial.

Introduction

Body size is among the most important traits of an organism (Bell, 2014). It influences, amongst other things, an organism’s potential diet range (Sinclair, Mduma & Brashares, 2003), the habitats it may occupy, its energy requirements (Oksanen et al., 1981), its ability to defend against predation (Roff, 1992), its development (Gillooly et al., 2002) and viable reproductive strategy (Tuomi, 1980). As such, a great deal of effort has been put into understanding the patterns and processes in body size evolution and how this varies between clades, through time and between different ecological groups.

In a recent paper, Reisz & Fröbisch (2014) examined body-size evolution in the earliest terrestrial herbivorous vertebrate. During the first establishment of terrestrial ecosystems in the Carboniferous and Early Permian, high-fibre herbivory appeared independently in at least four different lineages: Edaphosauridae and Caseidae from the synapsid (mammal-line) amniotes, Captorhinidae from the sauropsid (reptile/bird-line) amniotes and Diadectidae from the stem-amniote diadectomorph lineage. Reisz & Fröbisch (2014) noted that, in these four families, the evolution of herbivory appeared to be consistently correlated with increased body size. The earliest members of each of these four clades are considered to be small carnivores or insectivores, and the herbivorous members of these clades appeared in each case to be noticeably larger than their carnivorous ancestors. Moreover, they suggested that there was more pronounced increase in body size in the herbivorous members of these clades than in closely related carnivores.

An association between herbivory and large body size has been noticed in other clades e.g. mammals (Gaulin, 1979; Fleming, 1991; Isbell, 1998; Price & Hopkins, 2015), birds (Morton, 1978; Dudley & Vermeij, 1992; Klasing, 1998) and lizards (Pough, 1973; Schluter, 1984; Cooper & Vitt, 2002). Two possible explanations have been offered for the association. The first is based on the Jarman-Bell Principle (Geist, 1974), originally used to explain body size evolution in ungulates but since applied to other clades (e.g. Gaulin, 1979; Fleming, 1991; Isbell, 1998). This principle posits that, since smaller animals have higher metabolic energy requirements relative to their body size, smaller herbivores are limited to more easily digestable plant material such as roots and fruits. Larger members of a herbivorous clade have lower energy requirements relative to their body size, and so are able to subsist on less digestable plant material such as leaves. Since less digestable plant material is more abundant, those able to subsist on it (the larger herbviores) have a selective advantage over their smaller relatives.

The second explanation for the association between large body size and herbivory, dubbed the abundance-packet size hypothesis by Olsen (2015), is based on the absolute nutrient requirements rather than nutrient requirements relative to body size. A lineage which increases in mass will require larger amounts of food. Therefore, the lineage must either feed on larger prey (macro-carnivory) or find a more abundant food source such as plant material. It should be noted that the abundance-packet size hypothesis differs from the Jarman-Bell Principle in the proximal cause for the association between herbivory and larger body size; the Jarman-Bell principle posits that in a herbivorous lineage there will be a selective pressure towards larger body size, whereas the abundance-packet size hypothesis posits that in a lineage with a large body size there will be a selective pressure to a more herbivorous diet.

Reisz & Fröbisch (2014) did not quantitatively test their hypothesis that the earliest herbivores showed a greater trend towards larger body size than the carnivores; rather, it was tentatively suggested based on a visual examination of plotting diet and body size over a phylogeny. However, such inferences do need to be rigorously tested; for example, a more detailed examination of body-size evolution in Therizinosaurs showed that previous assumptions of a trend towards large body size in these herbivorous theropods was unfounded (Zanno & Makivicky, 2012).

Of the four clades examined by Reisz & Fröbisch (2014), the captorhinids provide the best case study for testing their theory. They are the longest lived of the clades, surviving from the late Carboniferous until the end of the Permian. They are also the most speciose, with more than 25 species currently described. Moreover, a recently published and comprehensive (19 captorhinid terminal taxa) phylogeny exists, well resolved and with reasonably high node supports (Reisz et al., 2015), providing an excellent framework for an analysis of body-mass evolution.

The captorhinids were the most diverse sauropsid clade in the Paleozoic. They first appear in the fossil record in the Virgilian aged Hamilton Quarry of Kansas (Müller & Reisz, 2005) before diversifying during the early Permian. By the Middle Permian they had achieved a global distribution, being known from North America, Europe, Asia and Africa. The first herbivorous members of this clade appeared during the Kungurian (Doddick & Modesto, 1995). Crucial to the evolution of herbivory in captorhinids were the multiple rows of maxilliary and dentary teeth. This feature first appeared in the insectivore/omnivore Captorhinus (Doddick & Modesto, 1995) but in the high-fibre herbivorous taxa up to 11 rows of bullet-shaped maxilliary teeth, combined with a propalineal motion of the lower jaw, form an effective grinding and shredding surface for processing plant material (Modesto et al., 2007).

Here I examine changes in rate and mode of the evolution of size in captorhinids. In particular the evolution of the carnivorous and herbivorous members is compared, in order to test the hypothesis of Reisz & Frobisch. The applicability of the Jarman-Bell Principle and the abundance-packet size hypothesis to captorhinid evolution is also examined.

Materials and Methods

Proxy for body size

The estimation of body mass in extinct organisms is unsurprisingly difficult, in the absence of a complete skeleton and soft tissue. Some workers have attempted volumetric reconstructions (e.g. Colbert, 1962; Hurlburt, 1999; Motani, 2001), but these require relatively complete skeletons and are not useful in examinations of body mass spanning entire clades. Most other estimates have used a single measurement as a proxy for mass e.g. dorsal centrum cross section (Romer & Price, 1940), humerus and femur shaft circumference (Campione & Evans, 2012). This of course requires those taxa not possessing the necessary elements to be ignored, but in large analyses (e.g. Benson et al., 2014) on dinosaurs, a small number of deletions should not mask the overall pattern.

Since Captorhinidae are a small clade (18 taxa included in the most comprehensive phylogenetic analysis) useful proxies for body size are limited by the available material. Due to the fact that all terminal taxa analysed by Reisz et al. (2015) possess skull material, but only a limited number have postcranial material preserved, it was decided to use the skull length as a proxy for size. While cranial material is not often used in calculations of body mass, a precise mass in grams is not necessary for the analyses herein; an estimate of relative difference in size is the most important. It is obviously possible for skull size to vary relative to the rest of the body, but comparison of skull measurements to postcranial measurements for those taxa which preserve both indicate that, in captorhinids at least, this does not appear to be a serious concern (see measurements provided by Reisz & Fröbisch, 2014). Using skull material allows the study to be as comprehensive as possible. Only two taxa are not represented by a skull complete enough to measure the length (Captorhinikos valensis and Gansurhinus quingtoushanensis), and both of these do preserve skull material, so an estimate of skull length could be obtained by comparing the length of elements preserved in these taxa to the length of elements preserved in closely related taxa. The skull lengths were log transformed prior to analysis (see Supplementary Materials for skull-length data).

Phylogeny and time calibration

The most recent and comprehensive phylogeny of captorhinids (Reisz et al., 2015) formed the basis of the analyses presented herein. The phylogeny contains 19 captorhinid taxa, of which 6 are considered high-fibre herbivores (Fig. 1). The phylogeny was time calibrated using the method of Brusatte et al. (2008) in the R 3.03 (R Core Team, 2014) package paleotree (Bapst, 2012); zero-length branches resulting from inconsistencies between the order of branching and the order of tip appearance in the fossil record were eliminated by sharing the zero-length branches equally along the non-zero-length branch immediately ancestral to them. While other time calibration methods are available for use on extinct datasets (for summary see Bapst, 2014a; Bapst, 2014b), the use of the Brusatte et al. method is less subjective than the addition of arbitrary amounts of time to zero-length branches. Meanwhile, methods like the Cal3 method (Bapst, 2013) are not suitable for datasets with poor sample size and low resolution. After time calibration, the non-captorhinid outgroups (Paleothyris and Protorothyris) were dropped.

Figure 1 The phylogeny of Captorhinidae, illustrating the evolution of high-fibre herbivory.

Two of the 100 time calibrated phylogenies used in this analysis. The thick branches indicate the observed range for non-singleton taxa. The tip labels in green indicate those taxa inferred to have a high-fibre herbivorous diet. The pie charts represent the probability of each dietary regime inferred for each node, deduced by maximum likelihood ancestral state reconstruction using the ace function in the R package ape. (A) MPT 1: Opisthodontosaurus is the sister to the clade containing Rhiodenticulatus and all captrorhinids more derived. (B) MPT 2: Opisthodontosaurus is the sister to Concordia.

To resolve the uncertainty surrounding the age ranges of taxa, 100 time calibrated trees were generated using the method of Pol & Norrell (2006). For each tree, the ages of each taxon were drawn at random from a uniform distribution of the full possible age range. A single age was drawn for singletons, a first and last appearance for taxa represented by more than one specimen. Subsequent analyses were performed on all 100 trees to assess the impact of uncertain age ranges (see Supplementary Materials for trees in nexus format). Since the (Reisz et al., 2015) phylogenetic analysis found two most parsimonious trees (MPTs, differing in the position of Opisthodontosaurus), 50 of these 100 trees were based on one MPT, 50 the other.

Models of rate and mode of evolution

Model fitting

When examining the evolution of a continuous trait such as body size and comparing it in different clades, ecological groups or time periods, one must consider both the rate and the mode of evolution. Models such as Brownian motion (BM) assume evolution via a statistically random walk with a constant normally distributed deviate from the observed morphology. In macroevolutionary processes, this can result from randomly varying selection a lack of selective pressure in any particular direction and a lack of variation in rate (Mooers, Vamosi & Schluter, 1999; Pagel, 1997; Pagel, 1999). As such, a clade evolving by simple Brownian motion will show no directional trend in trait mean, but instead the trait variance will increase through time. More complicated models can add parameters to provide a more detailed simulation of evolution. One can, for example, incorporate a directional trend to the BM model; the trait variance will still increase through time, but the mean will either increase or decrease. The Ornstein Uhlenbeck (OU) model incorporates an adaptive optimum to which trait values are drawn; the further a lineage strays from this optimum, the more strongly it is drawn back (Hansen, 1997). Once the trait has reached the adaptive optimum, it will show a constant variance and mean through time. Rate variation has also been examined, such as in the early burst (EB) model (Harmon et al., 2010), where the rate of change decreases exponentially from an initial maximum, causing the increase in trait variance to be rapid in the early history of a clade, but to then slow. Further models have been developed allowing shifts in either rate or mode of evolution between clades (O’Meara et al., 2006; Thomas, Meiri & Phillimore, 2009) or at specific points in time (Slater, 2013).

Maximum likelihood was used to fit three sets of models to the observed size estimates, with the Akaike weights used to deduce which model in each category fits best. The three categories represent 1) models of the evolution of the continuous trait (body size) alone; 2) models of the co-evolution of body size with a dietary regime; 3) models of the evolution of body size subjected to external influences at specific times.

The models in the first category included Brownian motion, Brownian motion with trend, Ornstein Uhlenbeck, the TM1 model and the SURFACE model. Under the TM1 model the trait evolves by Brownian motion, but one or more heritable shifts in rate may occur at any node. If a shift occurs at a node, an increase or decrease in rate is deduced for all lineages descended from that node (Thomas, Meiri & Phillimore, 2009). Under the SURFACE model, the trait evolves under an OU process, but one or more shifts in adaptive peak may occur at any node; that is, the descendants from the node at which the shift occurs will be drawn to a different trait value (Ingram & Mahler, 2013). These five models represent models where captorhinid body size evolution is independent of any specified factor; evolution is either consistent across captorhinids, or can shift but at entirely unspecified points. The BM and BM with trend models were fit using the fitContinuous function in the R package Geiger (Harmon et al., 2008); the OU model using the functions in the package OUwie (Beaulieu et al., 2012); the TM1 model using the transformPhylo.ML function in the package motmot (Thomas, Meiri & Phillimore, 2009); the SURFACE model using the runSurface function in the package surface (Ingram & Mahler, 2013).

The models in the second category allow different rates or modes of evolution to occur under different dietary regimes. Three such models are tested: BM-V OU-M and OU-MV. Under the BM-V model, body size evolves by Brownian motion but with rates of evolution differing between the carnivorous lineages and the herbivorous lineages. The OU-M model represents body size evolution under an OU process, but with the different dietary regimes drawn to different adaptive optima. The OU-MV model is similar to the OU-M model, but allows a change in rate of evolution as well as adaptive optimum under different dietary regimes. All three of these models were fit using the functions in the R package OUwie.

The third category of models tests for the possibility of extrinsic influences on the evolution of body size in captorhinids; that is, changes in rate or mode are related to a change in the organism’s environment (both biotic and abiotic) rather than any evolutionary innovation within the clade itself. Were this to be the case, one would expect a shift in rate or mode to occur at a specific point in time, and affect all lineages after this point, rather than affecting all taxa descended from a specific node. The captorhinids evolved at a time of great changes in environment. Throughout the late Carboniferous and Permian, there was a trend towards a warmer, dryer climate (Rees et al., 2002). At the end of the Carboniferous, there was a collapse in the equatorial rainforest and a shift towards seasonally dry climates (DiMichelle et al., 2006; DiMichelle et al., 2009). This coincided with, and possibly caused, a radiation in amniotes (Sahney, Benton & Falcon-Lang, 2010). In the Early Permian, during the Sakmarian stage, there was an abrupt shift towards higher temperatures and accelerated deglaciation (Montanez et al., 2007), possibly coinciding with a brief drop in amniote diversity (Brocklehurst et al., 2013). During the Kungurian and Roadian there was transition from an early Permian fauna dominated by pelycosaurian-grade synapsids and abundant amphibians to a Middle Permian fauna dominated by therapsids, possibly accompanied by a mass extinction event (Sahney & Benton, 2008; Benson & Upchurch, 2013; Brocklehurst et al., 2013; Brocklehurst et al., in press).

Slater (2013) described models to test the possibility of shifts in rate or mode coinciding with a specific point in time. The Rate Shift (RS) model assumes evolution by Brownian motion, but with a shift in rate at a specified point in time. The Ecological Release (ER) model allows a shift in mode at the specified time from OU to BM. Both of these models were tested four times, each with a different time specified for the shift to occur. The four times were the late Gzhelian (coinciding with the rainforest collapse and amniote radiation), the end of the Sakmarian (coinciding with the temperature spike), the early Kungurian and early Roadian (the time of the tetrapod faunal turnover and Olson’s extinction).

Having found the best fitting model in each of these three categories, these three models were compared to find a single model which overall best fits the evolution of body size in captorhinids.

Rate variation

When considering the evolution of a trait such as size, one cannot only consider shifts in the rate. Increasing the rate of evolution under a BM model increases the rate of evolution in both directions, towards larger and smaller. In order to examine whether there is indeed a greater tendency towards larger body size in the herbivorous captorhinids, one must ascertain: 1) whether rates of increase in size of herbivores are greater than rates of increase in carnivores; 2) whether rates of decrease in size of herbivores are less than rates of decrease in carnivores; and 3) whether rates of increase in size of herbivores are greater than rates of decrease. Such an examination requires a method which can assess rate variation along every branch in the phylogeny. The method of Venditti, Meade & Pagel (2011) was used to assess rate variation across every branch of the phylogeny. An MCMC analysis was carried out in BayesTraits V2.0 to calculate the pattern of rate variation which best fits the body size data to the time calibrated phylogeny. BayesTraits also scales the branch lengths of the phylogeny to represent rate variation. This method has an advantage over similar methods (e.g. Mooers, Vamosi & Schluter, 1999) in that it allows the scaling of not only individual branches, but the equal scaling of all branches within an entire clade, thus taking into account the possibility of rate heritability.

Along with the rate values calculated using the (Venditti, Meade & Pagel, 2011) method, each branch was assigned an inferred diet based on likelihood ancestral state reconstruction, and a direction of evolution (increase or decrease in size). The direction was deduced from ancestral state reconstruction of size, assuming evolution by BM but rescaling the branch lengths to represent the rate variation calculated in BayesTraits. The rates of both increase and decrease in body size in carnivores and herbivores were compared using the Mann Witney U test, calculated in R.

Results

The model from the first category (models of evolution independent of diet or extrinsic events) which best fits the body size data and phylogeny of captorhinids is the TM1 model representing a rate shift at a specific node in the tree (Fig. 2A). This model has a median Akaike weight score of 0.94. In all but three of the 100 time calibrated trees this model has a higher Akaike weights score than all the others, and in 83 the Akaike weight score is above 0.8. The majority of analyses (91) suggest this rate shift was an increase occurring at the same node: the clade containing Captorhinus, Captorhinikos valensis, Labidosaurus and the Moradisaurinae (Fig. 3), although there are a minority where the rate increase is found to have occurred only in the genus Captorhinus. This indicates the uncertainty surrounding the ages of taxa is influencing the results.

Figure 2 The fit of models of body size evolution to the phylogeny of Captorhinidae.

Boxplots showing the distribution of 100 Akaike weight values calculated for each model representing the fit of each model of body size evolution to the 100 time calibrated phylogenies. (A) Category 1 models (Evolution of body size alone). (B) Category 2 models (coevolution of body size with herbivory. (C) Category 3 models (evolution of body size with shifts in rate or mode at specified points in time). (D) Comparison of the best fitting models from each of the three categories. The abbreviation Gzh, Sak, Kun and Roa represent the time at which the shift in rate/mode occurred in the RS and EC models. Gzh: end of the Gzhelian; Sak: end of the Sakmarian; Kung: beginning of the Kungurian; Road: beginning of the Roadian. Other abbreviations as in the main text.

Figure 3 The shift in rate of body size evolution, identified by fitting the TM1 model to the phylogeny of Captorhinidae.

Two of the 100 time calibrated phylogenies of Captorhinidae, with the location of the rate increase indicated by the red branches. The branch lengths here represent the time until the first appearance of the taxa. (A) MPT 1. (B) MPT 2.

The best model from the second category (coevolution of body size and diet) which best fits the body size data and phylogeny of captorhinids is, in all 100 of the time calibrated phylogenies, the OU-M model (Fig. 2B). In all 100, it is found that the herbivorous lineages have are being drawn to a higher adaptive peak of body size than the carnivorous lineages. This model receives an Akaike weight score of above 0.8 in 78 of the tested phylogenies, and has a median Akaike wight score of 0.84.

The best model from the third category (coevolution of body size and diet) is the Rate shift model, with a rate increase occurring at the end of the Sakmarian stage (Fig. 2C). The median Akaike weights score of this model is only 0.63, indicating greater uncertainty when choosing between these models. There are 6 of the 100 time calibrated trees where a rate shift at the end of the Sakmarian fits the body size data less well than an ecological release (a shift from evolution under an OU process to a BM process) during the Kungurian or Roadian.

When the three best fitting models from each category are compared, it is the TM1 model with a rate increase at the node indicated in Fig. 3 which is overall found to best fit the captorhinid body size data, with a median Akaike weights score of 0.85 (Fig. 2D). It should be noted that this median Akaike weights score, while high, is not overwhelming. Only in 59 of the 100 time calibrated trees is the score over 0.8, and it is over 0.9 in only 37. The OU-M model received the second highest median Akaike weight score of 0.14%, but receives a score of above 50% in 14 of the 100 time calibrated trees (see Supplementary Data). This indicates that the uncertainty surrounding the ages of certain fossils is affecting the results. However, the majority (86) of the trees best fit a BM model with a rate shift at the clade indicated rather than a higher adaptive optimum for herbivores (see Supplementary Data). In none of the 100 time calibrated phylogenies is the RS model with a rate shift at the end of the Sakmarian found to be a better fit the either of the other two.

The variable rates analysis indicates that the mean rate of size increase in herbivores is higher than that of carnivores, and the Mann Whitney U test suggests the difference is significant (Table 1). However, there are also herbivorous lineages which show a decrease in size, and the Mann Whitney U test suggests that the rate of decrease is also significantly faster in herbivores than carnivores (Table 1). The rates of increase in size of herbivores was found to be lower than rates of decrease (albeit not significantly), while in carnivores the reverse was found; rates of increase in size are higher than rates of decrease, although again, not significantly (Table 1).

Table 1 Results of Mann Witney U tests.

Values of W and p-values resulting from the Mann Whitney U test comparing rates of lineages evolving in different directions and under different dietary regimes.

Comparison of rates	Sample size	Median of rates	W	P-value	
Herbivorous branches decreasing in size vs Carnivorous branches decreasing in size	3 vs 9	4.144 vs 1.234	25	0.036	
Herbivorous branches increasing in size vs Carnivorous branches increasing in size	9 vs 15	3.674 vs 1.308	104	0.029	
Herbivorous branches increasing in size vs Herbivorous branches decreasing in size	9 vs 3	3.674 vs 4.144	22	0.146	
Carnivorous branches increasing in size vs Carnivorous branches decreasing in size	9 vs 15	1.308 vs 1.234	52	0.379	

Discussion

Reisz & Frobisch (2014) put forward two theories about body size evolution in early herbivores. First, they suggested that herbivorous clades showed a trend towards increased body size. Second they suggested that this increase was more pronounced in herbivores than in closely related carnivores. Support for these hypotheses depends on observing one or more of these three possible results: 1) the Ornstein Uhlenbeck model with a variable trait optimum would be the evolutionary model best fitting the size data, and the trait optimum would be higher in herbivores than in carnivores; 2) the herbivorous branches which show an increase in size would have a faster rate of change than the carnivorous branches showing an increase; 3) the herbivorous branches which show a decrease in size should have a slower rate of change than the carnivorous branches showing a decrease.

Support for a variable optimum OU model being the best fitting is equivocal due to the uncertainty surrounding the age ranges. In most of the 100 sets of ages tested it is not the best supported model. In fact, the best supported model for size evolution is the TM1 model incorporating a shift towards higher rates of size evolution, and this shift does not coincide with the evolution of herbivory. Instead it occurred earlier, probably during the Sakmarian or Artinskian. The node at which this shift is inferred to have occurred does contain the herbivorous members of Captorhinidae, but also includes three species of Captorhinus and Labidosaurus meachami (Fig. 3), neither of which is considered to be a high-fibre herbivore (Dodick & Modesto, 1995; Modesto et al., 2007).

The results of the variable rates analysis in BayesTraits may be represented as a heat map (Fig. 4), in which high rates are represented by hot colours (purple and red). The results indicate an increase in rates of size evolution at the same node identified by the model fitting analysis. The greatest rate increase is identified in three tip branches: those leading to Captorhinus aguti, Captorhinus magnus and Captorhinikos valensis. Ancestral size reconstruction indicates that the size change on the branch leading to the herbivorous Captorhinikos valensis was an extremely rapid decrease (Fig. 5). Another herbivorous taxon which shows high rates of body size evolution (albeit not so high as Captorhinikos valensis) is Gansurhinus. Again, this herbivore is found to be experiencing a rapid decrease in body size.

Figure 4 Variation in rates of body size evolution within Captorhinidae, illustrated as a heat map.

Two of the 100 time calibrated phylogenies of Captorhinidae, illustrating variation in rates of body size evolution identified using the method of Venditti, Meade & Pagel (2011). The branch lengths here represent the time until the first appearance of the taxa. (A) MPT 1. (B) MPT 2.

Figure 5 The evolution of body size through time of the Captorhinidae.

Two of the 100 time calibrated phylogenies, illustrating both the age and inferred body size of each node. Ancestral body sizes are reconstructed using likelihood, assuming evolution by Brownian motion but scaling the branches to represent rate variation. Herbivorous lineages are coloured green. As the analyses do not take into account changes occurring within the observed ranges of the species, the observed ranges are here shown to experience no changes in body size. (A) MPT 1. (B) MPT 2.

Overall, the variable rates analysis also fails to support a tendency towards larger body size in herbivores. The rates of evolution along the herbivorous branches of the Captorhinidae are found to be, on average, higher than those of carnivores, but this increase in rate applies in both directions: both towards larger and smaller sizes (Fig. 6, Table 1). While this has resulted in comparatively large sizes in herbivorous taxa such as Moradisaurus, Rothianiscus and Labidosaurikos, all of which have skull lengths above 200 mm, extremely rapid rates of decreasing body size are also observed in herbivorous taxa such as Captorhinikos valensis and Gansurhinus (Fig. 5). Moreover, while herbivorous taxa do have a higher mean rate of skull size evolution than the carnivorous taxa, both the variable rates analysis and the model fitting analysis indicate that the shift in rate of evolution did not coincide with the evolution of high fibre herbivory, but instead occurred earlier in time and at a node containing both carnivorous and herbivorous taxa (Fig. 2). The results support neither a general trend towards larger size in herbivorous captorhinids nor an adaptive optimum of larger size; decreases in body size of high-fibre herbivores occur no less rapidly than increases. In fact, the results directly oppose Reisz & Frobisch’s hypothesis that there was a more pronounced trend towards increased body size in herbivores than in carnivores. The rate of size evolution in the high-fibre herbivores is found to be higher in branches which decrease in body size than in those which increase (Fig. 6), although the difference is not significant, possibly due to the low sample size (Table 1). Meanwhile the converse is found to be true in the carnivorous lineages; body size evolution is faster in lineages which show an increase (although the difference is lowest of those tested and is again not significant).

Figure 6 The evolution of multiple tooth rows in the Captorhinidae.

Two of the 100 time calibrated phylogenies used in this analysis. The tip labels in red indicate those taxa with multiple tooth rows. The pie charts represent the probability of an ancestral morphology including multiple tooth rows, deduced by maximum likelihood ancestral state reconstruction using the ace function in the R package ape. (A) MPT 1. (B) MPT 2.

These results allow the rejection of the Jarman-Bell Principle as governing body size in Captorhinidae. The Jarman-Bell Principle posits a selective pressure towards large body size within an already herbivorous lineage. Therefore, were it applicable to captorhinids, one would expect the OU-M model, with the herbivorous captorhinids being drawn to a higher body size than carnivores, to best fit the captorhinid data, and one would expect the herbivorous captorhinids to show higher rates of increase in body size than decrease. Neither of these predictions is borne out by the data. The OU-M model cannot be completely rejected due to the uncertainty surrounding the ages of taxa, but it is not the best fitting model in most cases. Meanwhile, herbivorous captorhinids show some extremely rapid decreases in body size, while most of the herbivorous lineages which increase in body size do so gradually (with the exception of the lineage leading to Labidosaurikos).

Rejection of the abundance-packet size hypothesis is more difficult. This hypothesis posits that lineages with a larger body size should experience a selective pressure towards a more herbivorous diet, but does not preclude the possibility of herbivorous lineages returning to a smaller body size. The transition to herbivory does appear to have occurred in lineages of above average size, which would support this hypothesis. One should note, however, the uncertainty surrounding how many transitions to herbivory there were and where they occurred (Fig. 1). It is unclear whether Captorhinikos chozaenesis represents a separate evolution of herbivory, or if Labidosaurus represents a reversal to a more omnivorous diet.

Since the evolution of herbivory may be rejected as the cause of this shift in rate of body-size evolution, an alternative explanation is necessary. Changes in rate and mode of evolution can either be intrinsic, relating to the evolution of a “key” morphological, behavioural or developmental innovation, or extrinsic, relating to a change in environment. An extrinsic cause would be supported if a shift in rate or mode occurred at a specific time rather than in a specific clade. This does not appear to be the case in the Captorhinidae; models involving a temporal shift in rate and mode fit the captorhinid phylogeny worse than the TM1 model. Therefore an intrinsic cause must be sought; one must consider the morphological variations within captorhinids.

One feature which characterises the more derived captorhinids is the increased efficiency of food processing. Multiple tooth rows have evolved at least twice; in Captorhinus aguti and in the clade containing Captorhinikos chozaensis and the Moradisaurinae, although the lack of this feature in Labidosaurus leads to uncertainty over the optimisation of this character (Fig. 6). The evolution of the propalineal motion of the lower jaw is another innovation which would improve food processing in this clade. The ability to perform such a motion has been suggested in Captorhinus (Heaton, 1979), Labidosaurus (Modesto et al., 2007) Captorhinikos valensis (Modesto, Lamb & Reisz, 2014) and the Moradisaurinae (Dodick & Modesto, 1995), a distribution which corresponds with the position of the inferred rate shift. One might reasonably suggest that the improved food processing could have widened the range of ecological niches available to captorhinids. Basal captorhinids were limited to a “grab and gulp” feeding habit, indicated by the simple conical teeth and suitable only for capturing small, non-resisting prey such as invertebrates (Hotton, Olson & Beerbower, 1997). The inferred evolution of propalineal jaw motion coincides with the node at which the rate shift occurred, and the majority of taxa descended from this node possess multiple tooth rows. The transition to a dentition and jaw morphology allowing processing of food after capture widens the diet available for both carnivorous and herbivorous forms; in carnivores it allows the capture of larger, more resistant prey and the detatchment of pieces for mechanical processing, while in herbivores it aids the digestion of tough fibrous plant material whilst reducing the need for gut fermentation inferred for more bulky herbivores such as caseids. One might therefore infer an “ecological release” as the increased food processing ability permitted captorhinids to explore as-yet untried ecological niches, increasing the rate body size evolution in both directions.

There are difficulties with inferring a causal relationship between the evolution of the specialised jaw motion and the rate shift. One must remember this evidence is purely circumstantial. Moreover the propalineal motion is in some cases difficult to infer; one can deduce its presence from wear patterns on the teeth (Heaton, 1979; de Ricqlés & Taquet, 1982), but such data is not always available and morphological correlates must be found. Dodick & Modesto (1995) suggested the increased length of the articulation between the lower jaw and the quadrate could be such an indicator, as could the vaulting of the skull roof which allows the required angle of adductor musculature. However, these lines of evidence conflict in the case of Captorhinus; the tooth-wear patterns suggest the ability to perform the jaw motion (Heaton, 1979), but Captorhinus lacks the supposed morphological correlates (Dodick & Modesto, 1995).

Despite these uncertainties a causal relationship between the evolution of improved oral processing equipment and increased rate of body size evolution is an extremely tempting one. Further confirmation could be provided by examination of other taxa with multiple rows of teeth and the specialised jaw motion. Baeotherates fortsillensis, for example, was not included in the analysis of Modesto, Lamb & Reisz (2014), on which that of Reisz et al. (2015) is based, due to the lack of material, but the single dentary preserved shows tooth morphology similar to Captorhinus aguti (May & Cifelli, 1998), and so could provide further information on the evolution of multiple tooth rows in non-herbivorous taxa. Meanwhile, taxa such as Gecatogomphius kavejevi, Kahneria seltini and Captorhinikos parvus would deliver further data on body size evolution in the herbivorous taxa. Modesto, Lamb & Reisz (2014) suggested further preparation of these taxa would be required before attempting to fit them in a phylogenetic analysis.

It is necessary to conclude with an acknowledgement that, while this study does cast doubt on the general hypothesis of Reisz & Frobisch (2014) that the earliest herbivores showed a pronounced trend towards larger body size, one should be careful about expanding the inferences presented here to other clades. Multiple clades evolved herbivory independently, and they all show great variation in size, tooth morphology and environmental preference. Nevertheless, this study strongly highlights the need for quantitative examinations of evolution. It is difficult to make inferences about evolutionary patterns and processes with visual examinations of trait data divided into coarse categories. In clades where a comprehensive, well-supported phylogeny exists, a great variety of tools are available to test such hypotheses and provide robust confirmation.

Supplemental Information

Supplemental Information 1 Time calibrated phylogenies of Captorhinidae.

The 100 time calibrated phylogenies used in the study, provided in nexus format. Brach lengths represent millions of years. The first 50 are based on MPT 1, the next 50 on MPT 2.

Click here for additional data file.

Supplemental Information 2 Raw data used in the study.

Data on skull length, dietary preference, presence or absence of multiple tooth rows, and age range of captorhinids used in this study.

Click here for additional data file.

Supplemental Information 3 Parameters of the TM1 model.

Parameters of the TM1 model (the best fitting of the category 1 models) when fitted to each of the 100 time calibrated phylogenies.

Click here for additional data file.

Supplemental Information 4 Parameters of the OU-M model.

Parameters of the OU-M model (the best fitting of the category 2 models) when fitted to each of the 100 time calibrated phylogenies.

Click here for additional data file.

Supplemental Information 5 Parameters of the RS-Sakmarian model.

Parameters of the Rate Shift model with a shift ath the end of the Sakmarian (the best fitting of the category 3 models) when fitted to each of the 100 time calibrated phylogenies.

Click here for additional data file.

I would like to thank the Fröbisch working group for their helpful discussion and support. Graeme Lloyd, Graham Slater, Johan Renaudie and Joanna Baker offered assistance with R and BayesTraits. Two anonymous reviewers gave many useful comments which greatly improved the paper.

Additional Information and Declarations

Competing Interests

Author Contributions

Data Deposition

The author declares that he has no competing interests.

Neil Brocklehurst conceived and designed the experiments, performed the experiments, analyzed the data, wrote the paper, prepared figures and/or tables, reviewed drafts of the paper.

The following information was supplied regarding data availability:

The research in this article did not generate any raw data.

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
