# Peer review of "Rates and modes of body size evolution in early carnivores and herbivores: a case study from Captorhinidae"

_PeerJ, doi:10.7717/peerj.1555_

## Round 0.1 · original submission · Major Revisions

· Academic Editor

Major Revisions

Dear Dr. Brocklehurst,

I have now received comments back from two reviewers. You will see that both reviewers consider the manuscript has potential and will be of interest to a broad audience, however in its present form some revisions are required. Particularly, both reviewers have highlighted some concerns about the methods, and some areas where the methods need further detail. Please pay careful attention to the request by Reviewer #1 to reword the description of model fitting for coherency, and, ideally, to redesign the method to fit more closely to the goal of the study (using OUwie). Reviewer #2 has requested further details on sampling of the phylogeny used to calculate rates, and has asked for all model parameters to be provided. I agree that both of these points are essential improvements that should be addressed. Finally, some suggestions are provided to improve Figure presentation.

Reviewer 1 ·

Basic reporting

The author sets out to test the hypothesis that body size is larger in herbivorous versus carnivorous captorhinids and that it attained this enlarged body size by evolving at a higher rate. Using several different phylogenetic comparative methods the authors conclude that a shift in rate of body size evolution does not coincide with the evolution of herbivory, but rather at a more basal node. Moreover, herbivorous captorhinids are not found to evolve at a higher rate than carnivorous captorhinids.
The topic is of wide interest and general importance to the field.

Experimental design

The authors tackle the topic with the latest phylogenetic comparative methods. However, the manner in which the methods are combined to produce the results lack a streamlined research design. In the end, results are not obtained with the best possible approach.

In the methods section phylogenetic comparative methods are described in a sequence that is not very coherent. First methods are described to fit homogenous rate models to body size across taxa. Then a model is described to fit a shift in rate of body size (TM1) across taxa. Then an approach is described to map diet onto body size evolution (OUwie). Then back to estimating shifts but across time rather than across taxa (RS, ER). And lastly back to shifts across taxa (Venditti’s reversible jump BM, or ‘rjBM’).

It would be advisable to properly differentiate between methods that estimate the structure and mode of evolution of a single variable (BM, single OU, EB, Trend, RS, ER, TM1, rjBM), and those that estimate the coevolutionary structure of the evolutionary landscape between a single variable (body size) and a mapped categorical variable (diet) (OUwie).

In order to optimally address whether body size in herbivores is indeed larger and has attained this larger body size at a higer rate relative to carnivores, methods that estimate the mode and structure of the evolutionary landscape should be incorporated into the method that estimates the structure of the evolutionary landscape across different regimes (OUwie).
This would considerably streamline the research design and allow for a more rigorous test.

Specifically, the author should translate the TM1 scenario (Figure 3) to a 2-regime OU hypothesis by mapping the clades that follow the rate shift as a different regime than those that precede it. It should be noted that the results of the rjBM are similar to those obtained by TM1 and would therefore result in the same 2-regime OU hypothesis.

The diet hypothesis presented in Figure 1 should also be translated in this manner.
Different models should then be fitted to these OU hypotheses. Each 2-regime hypothesis can then be used to test for a difference in body size and rate among the two regimes. Specifically, if “oumv” in OUwie indicated the highest statistical fit (aic weight >.9), the two regimes are indeed different in value and in rate. If on the other hand, “bms” indicates the highest fit, the two regimes are not different in value but they are different in rate. If “oum” indicates the highest fit, the two regimes indicate a different value but not a different rate. If “bm” indicates the highest fit, the two regimes do not indicate a different value nor a different rate.

Once the best fit model for the 2regime TM1/rjBM scenario is identified, it can be compared to the best fit model for the 2 regime diet scenario to ascertain the overall best fit evolutionary scenario. Together these results would provide a more rigorous test of the hypothesis using a unified research design.

EB, Trend, RS, and ER cannot be incorporated into this OU model testing design, but Figure 1 provides good reason why this is not a problem.

A good example of a recent paper that uses this approach:
Price SA & Hopkins SS (2015) The macroevolutionary relationship between diet and body mass across mammals. Biological Journal of the Linnean Society 115(1):173-184.

Validity of the findings

Results are stated clearly, but validity of the conclusions depends on the results of the above suggested research design.

Reviewer 2 ·

Basic reporting

Basic Reporting – Brocklehurst examines a small clade of fossil vertebrates to test recently proposed hypotheses regarding evolutionary tempo and mode in response to herbivory. Overall, I think there are some methodological errors that were made. There does seem to be evidence of a rate shift, but I think more should be done to account for the uncertainty in the phylogeny and time-scaling, and I am somewhat skeptical that the result would hold conclusively if this was done. Since the phylogeny is so small, it becomes much more important to show that the influential branches driving the signal are not simple artifacts of poor phylogeny estimation and time scaling. This was not done at all, and the method by which the 100 trees were sampled does not seem as though it would truly capture the range of possibilities given the dataset. A more complete exploration of this is required.

Parameters are not reported for any models. This is a major oversight and should be fixed with a table. Present AIC values, numbers of parameters of the model, and parameter estimates in a single table.

Experimental design

Experimental Design – It is critical in this paper (a comparison of rates) that the phylogeny is reasonably accurately measured. This is because errors in branch lengths have the potential to dramatically inflate or deflate rates, and will be especially be driven by taxa that have short branches. The description of how the phylogeny was constructed and time-calibrated requires considerably more detail. I'm very much confused as to how the age of nodes was estimated using time-calibration and very little detail is provided on this (besides how to treat “zero length branches”) . Consequently, not only age ranges of the tips should be sampled, but the age of the nodes should also be sampled.

It is also very confusing to me why for “non-singleton taxa, body sizes are assumed to be constant after their first appearance” when the models fit do not account for this (right?). If the models are Brownian for example, then how can you assume stasis for millions of years over the taxonomic duration of a species? I presume this is only done in the plot, and hopefully nothing odd is done in the model fitting. Would it be possible to include all the samples for a single species when they are present? This would provide very useful information for model-fitting and greatly increase your power.

Also, shouldn't the timing of the shift in the rate shift model be estimated? It seems a model with a rate shift in the early to mid Kungarian would be hard to distinguish from the TM1 model that was found to be the best fit.

Validity of the findings

Validity of Findings -
I am very confused as to how the p-values for the Mann-Whitney U tests can be as significant as they are. For example, the comparison of “Herbivorous taxa with increasing size” and “Herbivorous taxa with decreasing size”, if I look at figure 6, exactly 2 branches are colored green (herbivorous) and decrease. This is compared to 4 branches that increase. And yet the p-value is hugely significant. Also, the W values seem enormous. How was this calculated? Was this analysis done across all 100 tree? (In which case it would severely violate the assumption of independence across branches). It is also well known that rates often scale with the interval over which they are measured (See papers by Phil Gingerich 1983, 2001 etc). Therefore, assuming that this significance of the Mann-Whitney U test is significant, a further test is needed to make sure that this isn't just a product of different branch lengths. As I said earlier, artificially short branches can drive a lot of the signal in the dataset. (e.g. Captorhinus valensis)

Additional comments

Other comments

ln 180-181 – it seems pointless to cube the trait values before analyzing. This is equivalent to dividing all of the log-transformed traits by 3, and of course, will not affect the analysis other than tripling parameter estimates and their error. Linear size is as valid a trait to study as mass, and I don't see why it is important to perform this very rudimentary conversion.

ln 205 – Brownian Motion can arise as a by product of genetic drift, but it can also arise from, for example, randomly varying selection. This is more likely on macroevolutionary scales. BM in a statistical sense is simply any time that there is a constant normally distributed deviate for a given length of time, and as such, can have multiple biological explanations. Also, BM with a trend is still BM, so I think the language should be revised a bit to keep from defining BM as only being when there is no trend.

Ln 208 - “trait variance among lineage will increase linearly through time” would be more precise. In general, make sure you are communicating that the variance is among lineages, not with-population variance.

Ln 215 – “OU process will show a constant variance and mean through time” - Only true at stationarity

line 231 - “estimated” not “deduced”?

Ln 276 - “pattern” not “patter”

Figure 2 – Why not show a boxplot of Akaike weights for each of the 100 time-calibrated phylogenies than showing a rather uninformative barplot of the average values? As is, I don't the figure adds anything that a table couldn't.

Captions reversed for Figure 5 and 6

Table 1 - “Median” not “Media”. Also provide sample sizes.

Ln 333 – remove “heat map” from quotation marks.

Supplementary material – PLEASE do not provide nexus files in pdf format. This is very inconvenient to use. A plain text file would suffice.

Figure 5. Why have each boxplot alone in its own panel? Combine them all (or at least 1 x 2 panel for scaling purposes) into a single figure.

---

## Round 0.2 · accepted · Accept

· Academic Editor

Accept

Thank you for a careful revision that has closely followed the earlier recommendations of the reviewers.

A couple of very minor editorial points:

Please add a direct citation for use of the APE package (Paradis et al)
Ln76: “while” should be “whereas”
Ln 406: I would suggest to use an alternative word to “conjecturing” here
Ln 407: unclear what “this” refers to, please be specific

Reviewer 1 ·

Basic reporting

The authors have met all highlighted concerns. I consider this to be an excellent contribution.

Experimental design

The authors have met all highlighted concerns.

Validity of the findings

The authors have met all highlighted concerns.